# Neural effects of transcranial magnetic stimulation at the single-cell level

Maria C. Romero[1,2,3,4], Marco Davare[2,3,4], Marcelo Armendariz 🄳 [1,3] & Peter Janssen[1,3]

Transcranial magnetic stimulation (TMS) can non-invasively modulate neural activity in humans. Despite three decades of research, the spatial extent of the cortical area activated by TMS is still controversial. Moreover, how TMS interacts with task-related activity during motor behavior is unknown. Here, we applied single-pulse TMS over macaque parietal cortex while recording single-unit activity at various distances from the center of stimulation during grasping. The spatial extent of TMS-induced activation is remarkably restricted, affecting the spiking activity of single neurons in an area of cortex measuring less than 2 mm in diameter. In task-related neurons, TMS evokes a transient excitation followed by reduced activity, paralleled by a significantly longer grasping time. Furthermore, TMS-induced activity and task-related activity do not summate in single neurons. These results furnish crucial experimental evidence for the neural effects of TMS at the single-cell level and uncover the neural underpinnings of behavioral effects of TMS.

[1] Laboratorium voor Neuro- en Psychofysiologie, Katholieke Universiteit Leuven, Leuven, Belgium. [2] Onderzoeksgroep Bewegingscontrole & Neuroplasticiteit, Katholieke Universiteit Leuven, Leuven, Belgium. [3] Leuven Brain Institute, Katholieke Universiteit Leuven, Leuven, Belgium. [4] These authors contributed equally: Maria C. Romero, Marco Davare. Correspondence and requests for materials should be addressed to M.C.R. (email: mela.romeropita@kuleuven.be) or to M.D. (email: marco.davare@kuleuven.be)

A host of noninvasive brain stimulation (NIBS) techniques are widely used in neuroscience research[1–5]. The most attractive feature of these NIBS techniques is that they allow excitation or inhibition of neural tissue through the skull, so that causal inferences about the role of cortical areas can be made in healthy volunteers. Of all NIBS techniques, transcranial magnetic stimulation (TMS) has been successfully and widely used in volunteers and patients for over 30 years[5–11], but a number of fundamental questions remain unresolved. We know very little about the spread of the induced currents in the brain and their excitatory or inhibitory effects on neural activity at the single-cell level, both locally (i.e. immediately below the TMS coil) and at a distance (i.e., the response in interconnected brain areas). Inferences about the causal role of a cortical area in behavior critically depend on the ability to manipulate neural activity with high spatial selectivity, since interfering with multiple areas simultaneously could lead to erroneous conclusions. Selective spatial targeting also involves focusing on specific neuronal populations within a cortical area, but again, direct single-cell evidence for the effect of TMS on different neuronal subpopulations is lacking. Furthermore, numerous studies have demonstrated that the application of TMS can cause behavioral effects[12–18], yet how TMS-induced activity interacts with ongoing neural activity during behavior is entirely unknown.

Previous studies have employed computational modeling[19–23] or indirect measurements of the effect of TMS on neural activity, including behavior (the Motor Evoked Potential or MEP[24–28]), noninvasive electrophysiology (EEG[29–32]) or functional imaging[33–40]. Other studies have investigated the effect of repetitive TMS (rTMS) on hemodynamic, local field potential (ECoG) and single-cell responses in anesthetized animals[41–45]. However, none of these approaches provide the spatial and temporal resolution necessary to examine the impact of single-pulse TMS on individual neurons, the spatial extent of TMS effects, nor the interaction between TMS-induced and task-related activity during behavior. Recently, Mueller et al.[46] demonstrated the feasibility of measuring single-cell activity during single-pulse TMS, and Ortuño et al.[47] recorded extracellular activity in remote subcortical brain structures in awake behaving monkeys immediately after rTMS. The monkey model offers several important advantages for TMS studies because of its brain size, the pattern of sulci and gyri (which affects the current spread in the brain[19]) comparable to that of the human brain, and the possibility to measure single-cell activity during complex visuomotor tasks[48].

## Results

### Modeling of the electric field induced by TMS.
To study the TMS influence on neuronal activity, we performed model-based simulations of the induced electrical field and conducted combined single-pulse TMS and extracellular recordings in parietal area PFG during visually-guided grasping (VGG). First, we modelled the spatial spread of the TMS-induced current over the macaque parietal cortex using existing software (simNIBS[19]), simulating a distance of 15 mm between our TMS coil and the cortical surface (consistent with MR and CT imaging of both animals; Fig. 1a). According to the simNIBS model, TMS should induce a widespread activation in parietal cortex, extending to frontal and occipital cortex (Fig. 1b, left panel). In our single-cell experiments, we applied TMS at 120% of the resting Motor Threshold (rMT), which in Fig. 1b corresponds to the maximal electric field value (E-field, see online methods). To quantify the simulated TMS spread at this intensity (i.e., 120% rMT), we computed the E-field associated with the smallest effective TMS intensity capable of recruiting neural tissue (i.e., at 100% rMT, corresponding to 83% of the maximal E-field). Therefore, E-field

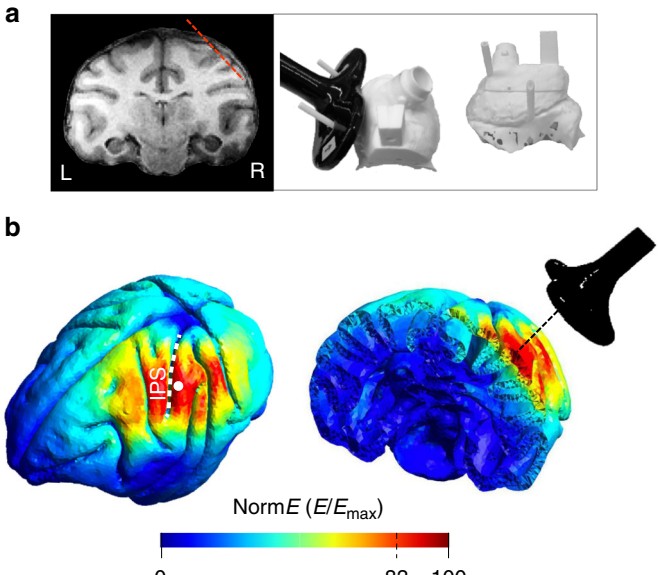

**Fig. 1** Brain targeting and model of the spatial spread. **a** Anatomical magnetic resonance image (left) and 3D models of the monkey's skull (right) indicating respectively, the trajectory of the electrode during recordings (red, dashed line) and the coil positioning method. The right panel shows a top view (left) and a lateral view (right) of a 3D model. During the experiments, a 25 mm figure-of-eight TMS coil (black) was rigidly anchored to the monkey's skull to allow precise and reproducible coil positioning across recording sessions. Two guiding rods were attached to the monkey's head implant based on MRI estimations of the cortical target coordinates. **b** TMS-induced electric field as modelled with simNIBS. Left: rotated model of the brain indicating the normalized electric field distribution (spatial spread) calculated for the D25 coil. The white dot indicates the center of stimulation (center of the coil). Right: Coronal section of the brain across PFG

values comprised between 83 and 100% can be used to define an area corresponding to the simulated spread of a TMS pulse (see Thielscher et al., 2015). With this procedure, the spread width measured ~20 mm both in the AP and medio-lateral direction, which is similar to the presumed spread in humans[49]. On a coronal rendering of the macaque brain (right panel in Fig. 1b), the simulation predicts an activation that is mainly limited to the cortex of the parietal convexity, excluding the region buried in the intraparietal sulcus.

### TMS effect on single neurons.
To investigate the neural effect of single-pulse TMS, we applied TMS pulses using a 55 mm coil (external diameter, Magstim coil model D25), which is smaller than the TMS coils traditionally used in humans (e.g., model D70), to take into account the smaller size of the monkey's skull. The coil was positioned over the parietal cortex while recording the spiking activity of neurons located under the center of stimulation (7 mm in the antero-posterior axis; 3 mm in the medio-lateral axis). We analyzed the spatial spread of TMS applied at different task epochs (light onset -the visual phase of the VGG task- and hand lift -the reaching phase of the VGG task-). Also, we applied TMS at two different intensities: 60% (low-intensity TMS) and 120% (high-intensity TMS) of the rMT, and recorded the activity of 538 parietal neurons (476 in the standard experiment and 62 in a control experiment; M1: 172, M2: 304) in two rhesus monkeys. To record as closely as possible to the skull, we precisely positioned the TMS coil as to target the parietal

convexity (area PFG[50]) based on anatomical MRI and CT imaging (Fig. 1a). Single-pulse TMS induced an artifact in the recordings lasting between 8 and 12 ms (Supplementary Fig. 1). Therefore, we excluded the epoch between 0 and 10 ms after TMS onset for all analyses.

Single-pulse TMS evoked a variety of effects on individual neurons in parietal cortex, as illustrated in Fig. 2. By far the most frequently encountered TMS effect was a short-latency burst of action potentials, starting in the first time bin after the TMS artifact (10 ms after the TMS pulse) and lasting less than 50 ms (Fig. 2a, left panels). The raster plot of the example neuron in the light-onset condition (top row) in Fig. 2a clearly illustrates that in the eight stimulation trials randomly interleaved with no-stimulation trials, TMS elicited a virtually identical burst of action potentials in the first 30 ms after TMS onset, with the earliest evoked spikes detected 11 ms after TMS onset. In the first three time bins (10–40 ms after TMS onset), the activity after high-intensity stimulation was significantly higher than in the no-stimulation condition (bin-by-bin analysis, two-sided Wilcoxon ranksum test; $p < 0.01$), whereas low-intensity stimulation did not induce any effect (Fig. 2a, right panels). The example neuron in Fig. 2a also illustrates that TMS applied at different time epochs of the task frequently induced similar responses, since the TMS-evoked response was virtually identical whether we applied TMS at light onset (top row) or at hand lift (bottom row in Fig. 2a, ANOVA, interaction between stimulation-no stimulation and trial epoch, $F = 0.04$, $p = 0.84$; df = 1). Single-pulse TMS did not exclusively induce excitatory effects in single neurons. The second example neuron (Fig. 2b) showed a complex pattern of TMS-induced activity. After an initial short-latency excitation (significantly elevated activity in the epoch 10–40 ms after TMS onset), the neuron became temporarily inhibited for approximately 50 ms, after which a second excitatory phase emerged lasting until 250 ms post-TMS onset. This excitation–inhibition–excitation pattern was observed when TMS was applied in either trial epoch (ANOVA, interaction between stimulation–no stimulation and trial epoch, $F = 0.27$, $p = 0.60$; df = 1). Overall, the example neuron in Fig. 2b clearly demonstrates that the application of TMS in awake behaving animals can induce both excitatory and inhibitory effects, which can reverberate through the cortical network for several hundreds of milliseconds. These two example neurons in Fig. 2 did not show task-related activity, but single-pulse TMS induced very similar effects in parietal neurons with task-related activity.

We performed two additional experiments to investigate the effect of coil orientation and stimulation intensity on single neurons. In 15 neurons showing a TMS effect, we inverted the coil so that the current flow was anterior-posterior (AP) instead of posterior-anterior (PA), and recorded the TMS-evoked activity under those conditions. Although we could elicit a TMS-evoked burst of activity with an inverted coil orientation, the magnitude of this response was significantly lower than that recorded with the standard orientation ($p = 0.04$, Wilcoxon, Supplementary Fig. 2), providing the first evidence that AP vs. PA coil orientations recruit underlying neural populations differently[51,52]. In addition, we tested the effect of an intermediate stimulation intensity (100% rMT) in 18 neurons. The TMS-evoked activity at 100% rMT differed significantly ($p = 0.01$, Wilcoxon) from the activity during stimulation at 60% rMT (Supplementary Fig. 3), and was even slightly higher than the activity at the standard intensity (120% rMT, $p = 0.04$, Wilcoxon). Importantly, these results indicate that our standard stimulation intensity was sufficiently above threshold to activate neurons in parietal cortex.

**Spatial extent of the TMS effect**. Out of 476 neurons recorded in the standard experiment in two monkeys, 132 (28%) were significantly affected by single-pulse TMS (Fig. 2c), i.e., showed at least one time bin with a significant difference between the TMS and the no-TMS conditions at $p < 0.01$ (two-sided Wilcoxon ranksum test; 35% in monkey Y, 23% in monkey P). Of those 132 neurons, a large majority (77%) showed facilitation (72% in monkey Y, 79% in monkey P), a smaller fraction (17%) showed inhibition (monkey Y: 16%, monkey P: 18%), and an even smaller fraction (7%) showed both facilitation and inhibition (monkey Y: 11%, monkey P: 3%; Fig. 2c). In a small fraction of the neurons (11%; monkey Y: 11%, monkey P: 10%), the magnitude of the TMS-induced response depended on the time of stimulation (ANOVA, significant interaction between the factors *epoch* and *stimulation*, $p < 0.05$).

The example neurons in Fig. 2a, b were recorded in the parietal convexity at the center of stimulation, immediately under the TMS coil (~15 mm under the skull), where the TMS-evoked response was maximal. However, to chart the spatial extent of the TMS effect, we also recorded at different distances from the center of stimulation with a spacing of 1 mm, both in the AP direction (seven grid positions in monkey Y; nine in monkey P) and in the medio-lateral direction (three grid positions in both monkeys) (Fig. 3a). The average population responses recorded at the five central positions in the grid (Fig. 3b, c) reveal the unexpected focality of single-pulse TMS effects. In monkey Y, two recording sites spaced a mere 1 mm apart showed a significant and transient increase in activity (two-sided Wilcoxon ranksum test comparing high stimulation and no stimulation trials; $p = 0.019$ and $0.014$), whereas the surrounding recording sites were barely affected by TMS (Fig. 3b). The recording site located 1 mm posterior to the center of stimulation also showed a weaker but significant effect (two-sided Wilcoxon ranksum test; $p = 0.016$). In the second monkey (Fig. 3c), we observed the main TMS-induced response in a single recording site (two-sided Wilcoxon ranksum test; $p = 0.006$), which was surrounded by recording sites showing no effect of TMS. The interaction between grid position and TMS was significant in a nested ANOVA analysis (with cells as a nested variable) in both animals ($p = 0.0002$ in monkey Y and $p = 0.0001$ in monkey P). In addition, we observed a second marginally significant (two-sided Wilcoxon ranksum test; $p = 0.038$) focus 3 mm away from the center (Supplementary Figs. 4 and 5). Figure 3b, c also shows that low-intensity single-pulse TMS had very weak effects in general, although we could detect excitatory effects of low-intensity TMS in individual neurons. The average population activity also illustrates that the TMS-evoked response was confined to the first 100 ms after TMS onset, although individual neurons occasionally showed TMS effects lasting up to 200 ms (Supplementary Figs. 4, 5 and 6).

The TMS pulse not only influenced the average firing rate of neurons under the center of the coil, but also induced oscillatory single-unit activity. We performed a spectral analysis on the single-unit activity using a Hanning-tapered Fourier transformation, and observed significantly higher low-frequency oscillatory activity after high-intensity TMS (compared to low-intensity TMS and no TMS, Supplementary Fig. 7). Interestingly, this increase in power in the lower frequencies (below 10 Hz) was also present outside the center of stimulation. Hence, single-pulse TMS induced low-frequency oscillatory activity across a region of cortex that was much wider than the area where we measured significant increases in single-unit firing rate.

We based the above analyses and plots on a compilation of all neurons recorded per grid position. However, we also noticed a high degree of spatial selectivity in the TMS-evoked response along the electrode penetration track, i.e., in the dorso-ventral direction, occurring around the center of stimulation. To

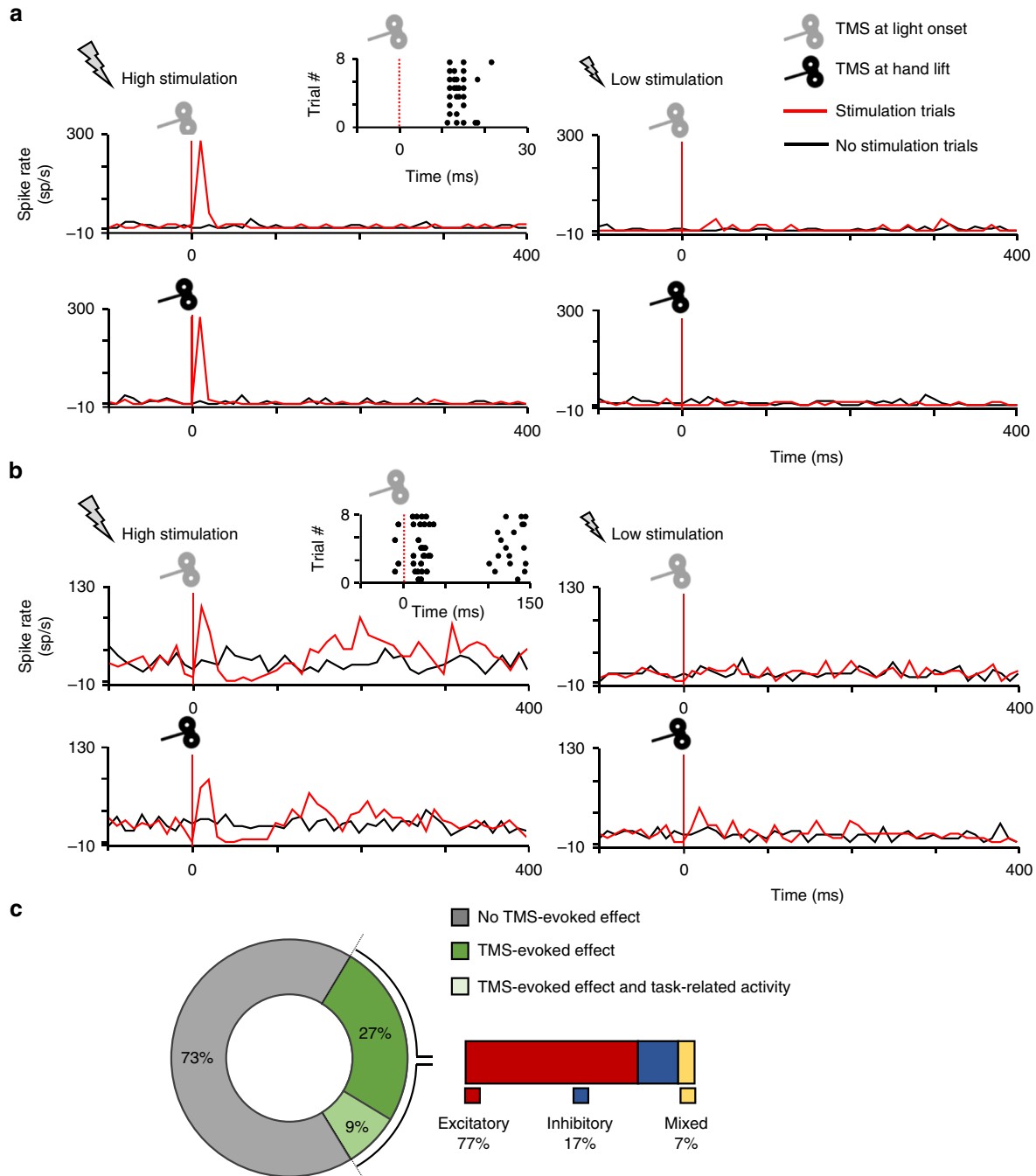

**Fig. 2** Combined single-pulse TMS and extracellular recordings in parietal area PFG. **a** Example neuron exhibiting a short-lasting excitatory response to high TMS stimulation (120% rMT; left panel) compared to low stimulation (60% of rMT; right panel). Stimulation (red line plots) and no stimulation trials (black line plots) were randomly interleaved during the task. In a vast majority of PFG neurons, single-pulse TMS evoked a short-lasting burst of activity, which emerged around 20 ms after stimulation, and lasted for approximately 40 ms. The induced excitation did not depend on the time of the stimulation, reaching similar amplitudes when the pulse was applied at 'light' onset (gray coil: visual epoch of the VGG task) or at 'hand lift' (black coil: reaching epoch of the VGG task). A detailed, trial-by-trial, illustration of the excitatory effect is shown in the raster plot. **b** Second example neuron exhibiting a combined excitatory-inhibitory pattern of TMS-evoked activity in response to high (left panel) but not low stimulation (right panel). After an initial excitatory response, we observed a longer-lasting neuromodulation phase combining both inhibitory and excitatory periods for at least 300 ms. Similarly to A, the TMS-induced excitation did not depend on whether TMS was applied during the visual (gray coil) or reaching (black coil) epoch of the task. **c**: Proportion of PFG recorded neurons with no TMS-evoked effect (gray), TMS-evoked effects (green) and both TMS-evoked and task-related neural responses (light green). The inset on the right shows that amongst cells with TMS-evoked effects, we predominantly found excitatory (red), and few inhibitory (blue) or mixed (yellow) TMS-induced neural responses

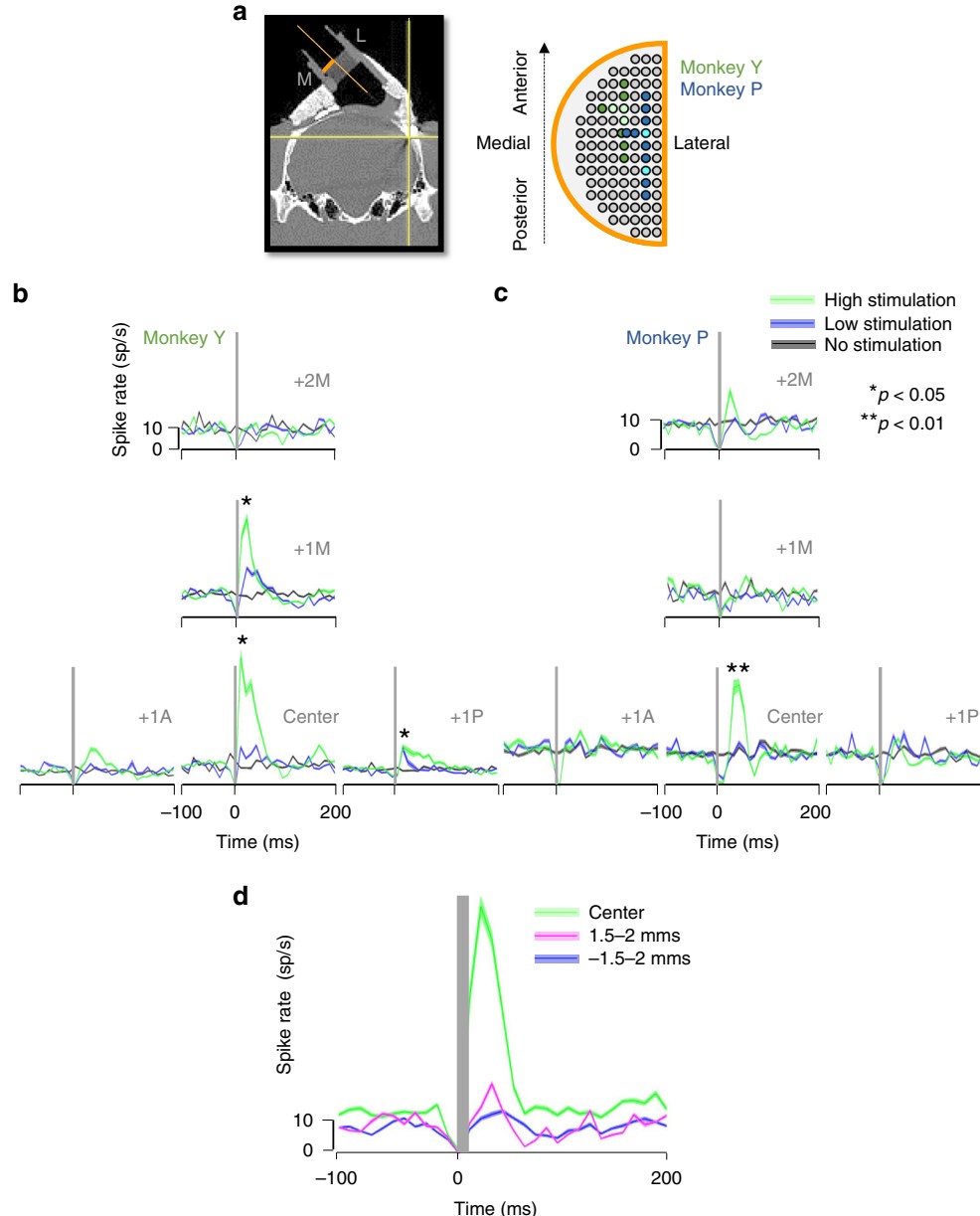

**Fig. 3** Spatial spread of the TMS-evoked activity. **a** Diagram of the recording grid (orange) used during the experiments. Colored electrode positions in the grid (green: monkey Y; blue: monkey P) indicate the extent of the area mapped in each animal. Brighter color shadings indicates grid points with significant TMS-evoked effects (i.e., hotspots). **b, c** Average spike rate obtained for all neurons recorded per grid position in response to high stimulation (green: 120% rMT), low stimulation (blue; 60% rMT) and no stimulation (black), when TMS was applied at 'light' onset. Graphs are arranged in a T-shape fashion in order to represent the activity maps spatially for grid positions located immediately under the center of the TMS coil (area mapped: from center to ±1 mm antero-posterior and + 2 mm medial). Asterisks indicate statistical strength (two-sided Wilcoxon ranksum test; *$p \leq 0.05$; **$p \leq 0.01$). B, Monkey Y and C, Monkey P. For this and the next panel, shading indicates ± the standard error. **d** Analysis of the spatial spread along the electrode penetration axis (dorso-ventrally and medio-laterally oriented). The CT scan (A, left panel) illustrates the trajectory of the electrode (orange dotted line). The line plots show the average spike rate observed for all neurons recorded at the same central grid position (aligned to the center of the coil) but at different electrode depths (green: 0 depth or center of stimulation; blue: 1.5–2 mms dorsal to the center of stimulation; magenta: 1.5–2 mms ventral to the center of stimulation) in response to high TMS stimulation (120% rMT). All averaged data were obtained for the TMS applied at 'light' onset

illustrate this observation, we first identified the depth at which the TMS-evoked response was maximal for every electrode penetration (considered as the center of stimulation; Fig. 3b, c). Next, we averaged the activity for all neurons recorded within 500 microns of this position (the center depth) and all neurons recorded 1.5–2 mm dorsal and 1.5–2 mm ventral to this position (Fig. 3d). The analysis of the dorso-ventral spread was possible in seven sessions in monkey Y (35 cells) and five sessions in monkey P (36 cells). While the TMS-induced response at the

center depth was very robust ($N = 25$ neurons, 64% showing a significant TMS effect), single-pulse TMS evoked barely any response at the neighboring depths. Our results estimate the spatial selectivity of the TMS effect as a volume of cortex measuring ~2 mm on a side, at least one order of magnitude smaller than the simulation results (see Fig. 1b and Supplementary Fig. 8). In this activated region of cortex, the electric field strength was above 95% of the maximum, instead of the predicted 83%. Figure 3d also illustrates that the magnitude of

the TMS-induced response is effectively underestimated in the plots of Fig. 3b, c, since the latter contain neurons recorded at different depths. In order to visualize both the average TMS-evoked responses and the variability across neurons, we plotted the net evoked activity of every cell recorded at 9 grid positions (0–3 mm anterior, posterior and medial to the center of stimulation) when we applied TMS at light onset (Supplementary Fig. 4). Note that not every neuron in the center was activated by single-pulse TMS. Neurons with a significant TMS effect showed an average transient increase in activity of 27 spikes/s in the interval from 10 to 80 ms after TMS onset.

Our recordings mainly consisted of very large action potentials that could be recorded for at least one hour and with an average signal-to-noise ratio of 3 or more, as in the example recording in Supplementary Fig. 9. However, we also observed significant and virtually identical TMS-induced responses in multi-unit recordings (signal-to-noise ratio of 3 or less; see Supplementary Fig. 9a, b). Thus, single-pulse TMS might activate many different neuron types in parietal cortex.

**TMS effect on task-related activity**. We also examined how TMS-evoked activity interacted with task-related activity in parietal cortex. In total, 41 neurons (16 in monkey Y and 25 in monkey P) showed significant TMS-induced responses (in the interval between 10 and 200 ms after TMS onset) and significant task-related activity (in the interval between 10 and 400 ms after the hand lift). The example neuron (Fig. 4a) and the normalized average activity plot (Fig. 4b, top panel) illustrate that the TMS-induced response was strong but limited to the first 100 ms after the hand lift, while task-related activity only emerged 200 ms after hand lift. Interestingly, the average normalized activity in TMS trials was significantly lower compared to no-TMS trials in the interval 300–700 ms after hand lift (two-sided Wilcoxon ranksum test; $p = 0.047$), which was not the case in the low stimulation trials ($p = 0.453$, Fig. 4b, lower panel). In total, 11 out of 41 neurons (27%) showed a significant reduction in task-related activity after TMS, which was on average 67% of the task-related activity in the absence of TMS. Thus, single-pulse TMS caused an initial burst of action potentials followed by a prolonged reduction in PFG activity during object grasping.

Since task-related activity appeared relatively late in the trial (in the later part of the transport phase of the hand and around the lift of the object) and the TMS-evoked response was short, we could not determine whether task-related activity and TMS-induced activity would linearly summate in PFG neurons. Therefore, we ran a control experiment in which we applied single-pulse TMS 400 ms after the hand lift, at a moment when the average task-related activity was high ($N = 62$). To assess the extent to which TMS-evoked activity and task-related activity summate, we aligned the activity to the onset of the TMS pulse, and compared the average activity in trials when TMS occurred at hand lift (TMS at Lift trials) to the activity when TMS occurred 400 ms after hand lift (TMS at Lift + 400 trials, Fig. 4c). Before the TMS pulse, the activity was higher in the 'TMS at Lift + 400' condition than in 'TMS at Lift' because task-related activity was stronger late in the trial. Conspicuously, the TMS-evoked response at 'TMS at Lift + 400' peaked around 40 ms after TMS onset but remained significantly lower compared to that observed at 'TMS at Lift' (two-sided Wilcoxon ranksum test; $p = 0.022$, Wilcoxon ranksum test), indicating that the TMS-evoked response did not summate with task-related activity. Later in the trial (40–260 ms after TMS onset), the activity difference reappeared (two-sided Wilcoxon ranksum test; $p < 0.001$). These observations strongly suggest that TMS-induced activity and task-related activity do not linearly summate in PFG neurons.

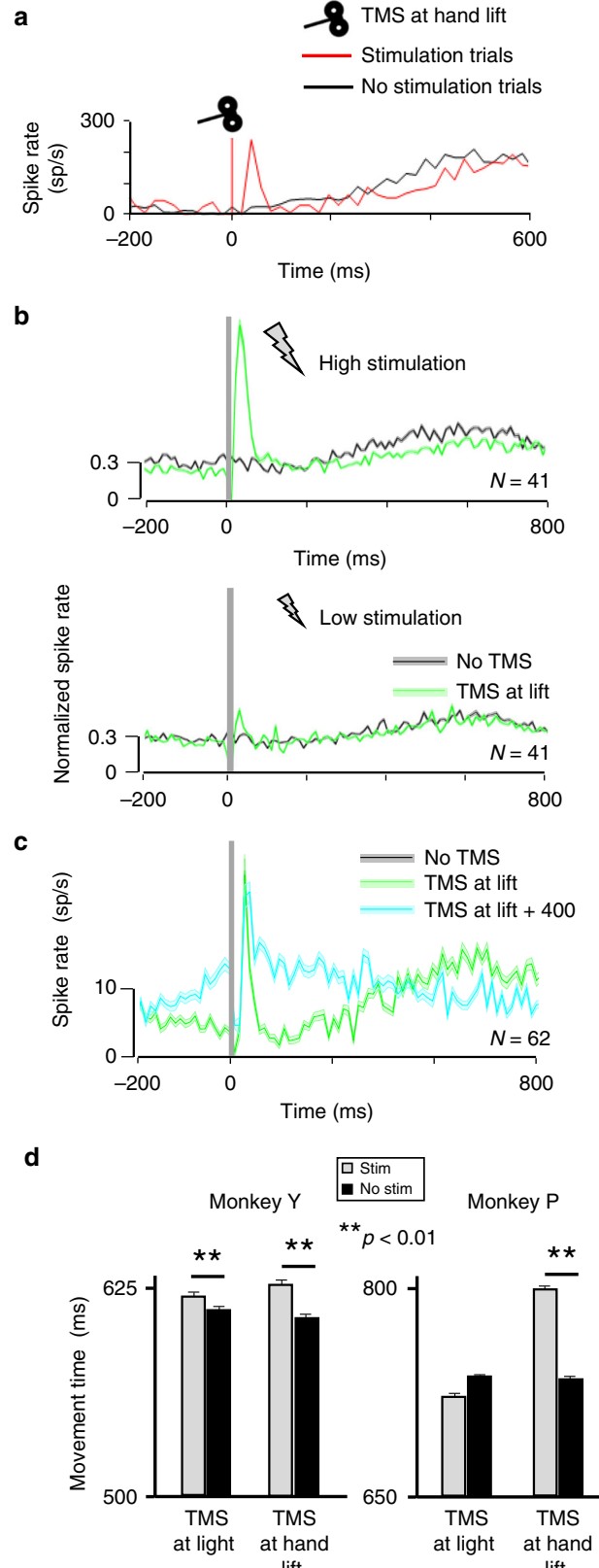

Importantly, while the above results show that single-pulse TMS interfered with task-related activity in parietal neurons (Fig. 4), we also found that single-pulse TMS caused significant behavioral effects in both animals. High-intensity TMS applied immediately after the hand lift significantly increased the time to grasp the object in both monkeys (Fig. 4d; a similar lengthening

**Fig. 4** Interaction between TMS effects, task-related activity and behavior. **a** Example neuron showing combined TMS and task-related activity (same conventions as in Fig. 2). The TMS-induced effect was, on average, similar to that described for non-task-related neurons, and did not overlap with the task-related activity. However, the TMS-induced excitation affected the total responsiveness of the neuron, which showed a global decrease in the firing rate during the reaching epoch (250 to 400 ms). **b** Normalized spike rate collected during the VGG task comparing the average activity observed at the center of stimulation for all cells showing both significant TMS-evoked effects and task-related activity in the two animals. All neurons were aligned at hand lift and compared across TMS intensities: high stimulation (green) versus no stimulation (black; higher panel) and low stimulation (green) versus no stimulation (black; lower panel). Shading represents ± the standard error. **c** Average spike rate obtained for all task-related neurons recorded in monkey P when TMS was applied at different timings during the reaching epoch (green: 'TMS at Lift'; cyan: TMS at Lift + 400'). The TMS-induced activity peak was similar across conditions. As for B, shading indicates ± the standard error. **d** Behavioral effect of TMS. The bar graphs show the total grasping movement times (time elapsed between the hand lift and object pulling) for both monkeys during TMS (gray bars) and no TMS trials (black bars). The effects of two different TMS timings are shown: TMS at light onset (visual epoch) and TMS at hand lift (reaching epoch). TMS applied during the reaching epoch significantly increased the grasping movement time in both monkeys. Asterisks indicate statistical strength (two-sided Wilcoxon ranksum test; $^{**}p \leq 0.01$)

of the total grasp time was present when TMS occurred after light onset, but only in monkey Y). We did not observe any TMS effect on reaction times, and low-intensity TMS had no effect on behavior (Fig. 4d). Thus, the prolonged reduction in neural activity in PFG after TMS onset was paralleled by an increase in the total grasp time.

## Discussion

Single-pulse TMS induced a highly localized and short-lived excitation of single neurons in parietal cortex of rhesus monkeys, which interfered with normal task-related activity. These findings have major implications for the interpretation of a very large number of TMS studies in human volunteers and patients.

The spatial selectivity of the TMS effect was unexpected based on previous studies modeling the electric field induced by TMS in humans (e.g., Opitz et al.[19]). We also simulated the electric field induced by TMS in the macaque brain based on state-of-the-art software (simNIBS), and found a predicted spread of at least 10 times broader compared to our single-cell results. However, TMS of primary motor cortex can elicit individual finger movements in humans[53] and in monkeys (our own observations). Near-threshold intracortical microstimulation studies in monkeys[54] estimate that the area in primary motor cortex for finger flexion and extension measures ~4 mm², which is remarkably consistent with our single-cell results. Thus, the focality of the TMS effect on individual neurons is in fact entirely consistent with previous behavioral results obtained with single-pulse TMS, especially in experiments where the TMS coil orientation is precisely aligned with the central sulcus for mapping different hand muscles[55]. Our data also confirm the limited penetration of TMS with a figure-of-eight coil. We measured robust TMS-evoked responses in the parietal convexity close to the skull, but very little if any effect further from the TMS coil, in more medial recording positions.

Our results may also be surprising in the light of numerous TMS-EEG studies, which showed oscillatory activity in the EEG induced by single-pulse TMS (e.g., Rosanova et al[56], Rogasch et al.[57], Fecchio et al.[58]). However, at least three possible factors could explain this apparent discrepancy. Firstly, the EEG signal

mainly reflects synaptic activations[59], whereas we performed extracellular recordings of action potentials in individual neurons. Secondly, we recorded the effect of single-pulse TMS in parietal cortex, and these parietal activations can certainly activate remote areas, e.g., in frontal cortex (as shown with electrical microstimulation during fMRI[60]), which could be detected by EEG. Finally, despite the absence of significant TMS-induced spiking responses at recording positions located 2 mm or more from the center of stimulation, TMS did induce strong oscillations (mainly in the low-frequency band) in the firing rate of these neurons, which may also contribute to the EEG signal. It is therefore worth noting that the spatial spread of TMS-induced spiking activity is clearly dissociable from TMS-induced oscillatory activity, which spreads more remotely. Overall, it is most likely impossible to identify the exact location of a cortical activation on a millimeter scale with EEG, as we did in our single-cell recording experiments. Thus, concurrent EEG-TMS recordings would have probably shown a TMS-evoked potential across a large part of parietal cortex. A similar reasoning may apply to the apparent discrepancy between our results and TMS-fMRI studies[36,61,62]. A previous study combining fMRI and single-cell recordings in parietal cortex of monkeys[63] has demonstrated that very restricted clusters of active neurons can drive an extensive fMRI activation.

The exquisite temporal resolution of single-cell recordings revealed that TMS induces a variety of effects in individual neurons in awake, behaving animals. While most neurons become activated within 40 ms of TMS onset, combinations of excitation and inhibition (the latter most likely caused by the activation of neighboring inhibitory interneurons slightly later in time) were not uncommon in our data set. We even observed secondary activations in recording sites further away from the center of stimulation. All these observations corroborate the idea that single-pulse TMS exerts robust effects on networks of neurons, which cannot be appreciated with population measurements such as fMRI, EEG or behavioral measurements.

Two technical factors may have influenced our findings. TMS may have induced a current in the microelectrode under the coil, causing direct electrical stimulation and consequently neuronal activation around the tip of the electrode. However, we did not observe such a neuronal activation at a distance of merely one millimeter from the center of stimulation (Fig. 3), although the microelectrode was undoubtedly still in the magnetic field under the TMS coil since we could record the artifact induced by TMS. Therefore, it is extremely unlikely that the TMS pulse induced electrical stimulation at the tip of the electrode. Secondly, the artifact in the recordings caused by the TMS pulse may have obscured neuronal responses at recording sites outside the center of stimulation. Indeed, we could not record any spikes in the first 8 ms after TMS onset. Nonetheless, the large majority of the neurons we recorded (89%) exhibited TMS effects that lasted at least 40 ms. Hence, we believe it is highly unlikely that we missed a substantial part of the neuronal responses induced by TMS. Finally, we used a smaller coil (D25; 55 mm external diameter) compared to the standard TMS coils used in human experiments (D70), in keeping with the proportionally smaller size of the monkey's skull. However, we obtained very similar intensity thresholds with the D70 and the D25 coils when testing individual finger movements in our monkeys (see "Methods").

We measured the effect of single-pulse TMS on the activity of single neurons generating large action potentials that we could record for an extended period of time (up to an hour) during an active motor task. These spikes most likely originated from large to medium-sized pyramidal neurons typical of area PFG[50]. However, TMS also influenced multi-unit activity characterized by small spikes, and—given the transient inhibition periods we

observed in neurons with large spikes—also of small inhibitory interneurons. In contrast, even at the center of stimulation, not every neuron recorded showed an effect of single-pulse TMS. This heterogeneity in the susceptibility of individual neurons to TMS pulses may be related to differences in the local orientation of neurons with respect to the induced electric field[19,64,65], which is difficult to estimate with our electrode approach. Thus, TMS influences a wide variety of cell types and evokes a variety of effects at the center of stimulation, the sum which is a transient excitation lasting less than 100 ms.

The effect of single-pulse TMS on the spiking activity of individual neurons we observed in awake behaving monkeys differed markedly from previous observations in anesthetized animals[66,67]. Notably, we did not observe long-lasting excitation (up to 300 ms after TMS onset) or suppression (up to two seconds after TMS onset), as reported by Moliadze and colleagues[66]. Undoubtedly, the absence of anesthesia and the active engagement in a motor task in our study could explain these differences. The active engagement in a task is also crucial for the similarity of our study to TMS studies carried out in human volunteers (e.g., Davare et al.[68] and Leib et al.[69]).

A unique advantage of our approach is that we could investigate the relationship between TMS-induced activity, task-related activity and behavior. TMS applied after the onset of the hand movement significantly prolonged the total object grasping time in both monkeys. Previous studies have demonstrated that neurons in area PFG are frequently active during object grasping[50] and during grasping observation[70], but no reversible inactivation of PFG during grasping has been published. In our experiments, task-related activity in PFG during the hand movement was suppressed after high-intensity single-pulse TMS. The increase in grasping time we observed may arise from this decreased task-related activity in PFG. Alternatively, the fast burst of action potentials immediately after TMS onset may have interfered with task-related activity in downstream cortical areas involved in grasping such as ventral premotor cortex, with which PFG is connected[60]. However, applying TMS at 400 ms after the hand lift (i.e., before the object was lifted) did not affect the grasping time, suggesting that the suppression of task-related activity following TMS may have been the most important factor. Our behavioral data strongly resemble previous findings in humans. TMS applied over the anterior IPS in human volunteers disrupts grasping in the presence[13] or absence[71] of a perturbation in object orientation requiring online adjustments. Additionally, TMS applied over the anterior IPS also prolongs reaching times when subjects have to correct for visual perturbations during the reaching movement[18,72]. Because the position of the TMS coil was fixed in our experiments, we could not stimulate control sites elsewhere in parietal cortex, but the single-cell data clearly showed that single-pulse TMS activated a patch of cortex in area PFG, at the parietal convexity. Thus, our study also represents the first causal evidence for area PFG playing a role in VGG using a reversible perturbation (for lesion studies see Faugier-Grimaud et al.[73] and Rushworth et al.[74]).

Single-pulse TMS during task-related activity could not excite the neurons above their normal activity levels (in the absence of TMS), indicating sublinear summation. This result is puzzling, since the average activity in our population of neurons may not have been maximal. Future studies will have to investigate the relationship between the neuronal 'state' and its excitability by TMS.

For several decades, NIBS techniques have proven to be major assets for both systems neuroscience[3,75,76] and the clinical practice[77–79]. Invasive animal studies combined with behavioral measurements can uncover new and unprecedented insights into the effects of brain stimulation on individual neurons and on

behavior, and will undoubtedly contribute to the refinement and development of novel stimulation protocols[44,80].

## Methods

**Subjects and surgical procedures.** Two male rhesus monkeys (Macaca mulatta; monkey Y, 12 kg; monkey P, 10 kg) were trained to sit in a primate chair. A head post (Crist Instruments) was then implanted on the skull with ceramic screws and dental acrylic. For this and all other surgical protocols, monkeys were kept under propofol anesthesia (10 mg/kg/h) and strict aseptic conditions. All experimental procedures were performed in accordance with the National Institutes of Health Guide for the Care and Use of Laboratory Animals and the EU Directive 2010/63/EU, and were approved by the Ethical Committee at KU Leuven. Intensive training in passive fixation and VGG began after 6 weeks of recovery. Once the monkeys had achieved an adequate level of performance, a craniotomy was made, guided by anatomical magnetic resonance imaging (MRI), over area PFG of the left hemisphere in monkey Y and the right hemisphere in monkey P. The recording chamber was implanted at an angle of approximately 45 deg with respect to the vertical, allowing oblique penetrations into the parietal convexity (Fig. 1a). To confirm the recording positions, glass capillaries were filled with a 2% copper sulfate solution and inserted into a recording grid at five different locations while structural MRI (slice thickness: 0.6 mm) was performed. During the experiments, a precise and highly reproducible positioning of the TMS coil was achieved by means of two guiding rods permanently attached to the skull using dental acrylic, oriented at an angle of approximately 90 deg with respect to the recording chamber. To estimate the position of the coil, we used anatomical MRI and computed tomography (CT scan) to build 3D printed models of the skull and implant of each animal (Fig. 1a, right panel). Thus, we visualized the relative position of the rods on the head, and calculated the distance between the coil and the brain when the coil was positioned over the rods touching the implant. Based on the CT-MR coregistered images, we calculated that the TMS coil was placed ~15 mm from the parietal convexity. The coil induced a posterior-anterior (PA) current over PFG.

**Modeling of the electric field induced by TMS.** A tetrahedral head model (mesh file) was created using simNIBS (simulation of Non-invasive Brain Stimulation Methods[23]) and a standard monkey brain template (National Institute of Mental Health Macaque Template: NMT[81]). The meshes consisted of five different volumes obtained from MRI images (T1; 0.6 mm isotropic resolution) by the progressive segmentation of five tissue types: white matter (WM), grey matter (GM), cerebro-spinal fluid (CSF), skull, and scalp[23]. The assigned conductivity values were fixed: 0.126 S/m (WM), 0.275 S/m (GM), 1.654 S/m (CSF), 0.01 S/m (skull), 0.465 S/m (scalp). Based on those volumes, simNIBS obtained surface reconstructions of the brain. Tissue segmentation was performed in a semi-automatic way using FSL tools (Functional MRI of the Brain Software Library[82]) which were manually corrected when required. The electric fields were calculated assuming a quasi-static regime[64,83], according to the following equation:

$$E = -\partial A/\partial t - \nabla\varphi$$

where $E$ is the electric field vector and $\varphi$ denotes the electric potential. The time derivative of the magnetic vector potential $A$ of the chosen TMS coil was given as input to the FEM calculations, and the rate of change of the coil current ($dI/dt$) was set to 1 A/μs. Our model was applied using two different coil files: a Magstim 70 mm figure-of-eight coil (D70) and a Magstim 55 mm (external diameter) figure-of-eight coil (D25; for animal use). The distance between the coil and the cortex was set to 15 mm, as measured in our MRI images, and the coil handle was oriented following the coordinates used in the actual experiment (coil centered on area PFG, projecting along the intraparietal sulcus). The simulations obtained from the two coil types produced similar results, although the total spread was smaller with the smallest coil size (Supplementary Fig. 10a). All parameters required for the modelling (change of coil current, distance to the cortex, conductivity values, center of stimulation) were identical across simulations. Using a custom-built measurement coil, we measured the relative output voltage obtained with both the 70 mm and the 55 mm figure-of-eight coils when stimulating at different intensities, taking readings at a 15 mm distance (see Supplementary Fig. 10b). Using this procedure, both coils showed similar output intensities (D25 coil 84% of the D70 coil at 70% of the maximum stimulator output intensity). Additionally, we measured the rMT with the two coils (D70 and D25) in two monkeys. The minimum intensity required to evoke finger twitches by applying single-pulse TMS on the contralateral primary motor cortex was very similar using the two coils (D70: ±40% and D25: ±35% of the maximum intensity for monkey 1; D70: ±46 and D25: ±46% of the maximum intensity for monkey 2).

**TMS stimulation.** TMS was applied at 60 and 120% of the rMT (see below for details) by means of a 55 mm figure-of-eight coil. We determined the rMT for each animal as the lowest stimulus intensity at which TMS of the primary motor cortex produced a contralateral finger or wrist movement, (right hemisphere on monkey Y and left hemisphere on monkey P) while the monkey held its hand in the resting position. In these experiments, the TMS coil was handheld over the primary motor cortex of the hemisphere contralateral to the recordings, at a distance of approximately 15 mm from the surface of the brain, similar to the distance in the

recording experiments. The parietal recording chamber was implanted but no electrode was inserted in the brain. In monkey Y, we used a Magstim 200 Stimulator (Magstim, UK), which delivered monophasic pulses (100 μs rise time, 1 ms duration, 120% rMT = 60% of max stimulator output), whereas in monkey P, we used a Magstim Rapid Stimulator (Magstim, UK), which delivered biphasic pulses (100 μs rise time, 250 μs duration; 120% rMT = 70% of max stimulator output). In a series of control experiments, we also used the Magstim Rapid Stimulator in monkey Y, and could reproduce the basic findings obtained with the Magstim 200. For the duration of the experiment, the coil was placed over the guiding rods tangentially to the skull. To provide additional stability, the TMS coil was held in place by means of an adjustable metal arm. Since our recording sessions averaged 4 h in duration, gel-filled cool packs were placed around the coil to prevent overheating (Nexcare; 3 M Company, Minnesota, USA).

**Single-cell recordings during single-pulse TMS.** During the application of single-pulse TMS, we recorded single-unit activity with tungsten microelectrodes (impedance: 1 MΩ at 1 kHz; FHC) inserted through the dura by means of a 23-gauge stainless steel guide tube and a hydraulic microdrive (FHC). Mimicking the artifact reduction strategy proposed by Mueller and colleagues[46], we used diodes and serial low-gain amplification to clip the artifact generated by the magnetic pulses and to prevent amplifier saturation. For this, we modified a regular BAK Electronics preamplifier (Model A-1) by connecting two leakage diodes (BAS45A) anti-parallel between the signal lines and ground before each stage of amplification. The initial front-end of the headstage remained unmodified to maintain the high-input impedance. Also, we covered the tungsten microelectrodes with polyamide tubing (TWPT-0079-30-50; Small Parts) to minimize vibration inside the guide tube. With these settings, the evoked TMS artifact in our signal ranged from 8 to 12 ms (Supplementary Fig. 2). Neural activity was amplified and filtered (300–5000 Hz) following a standard recording protocol for spike detection. Using a dual time-window discriminator (LabVIEW and custom-built software), we isolated individual neurons and the TMS artifact, which was detected online and subtracted from the neural data. In addition, we recorded the entire raw signal (after filtering) for the analysis of multi-unit activity. We monitored the right eye position using an infrared-based camera system (Eye Link II, SR Research, Ontario, Canada) sampling the pupil position at 500 Hz.

**Grasping task.** During the experiments, the monkeys were sitting upright with the head fixed. For each recording session, the hand ipsilateral to the recording chamber remained restrained within the chair, while the contralateral hand was placed on a resting device. A single grasping object (large sphere; diameter: 35 mm) was located in front of the monkey at a distance of 30 cm from the resting position. The resting position of the hand, the start of the reach to grasp movement and the pull of the object were detected by fiber-optic cables. Our motor task consisted of a visually guided grasping task (VGG), in which the monkey had to place the hand contralateral to the recorded hemisphere in the resting position in complete darkness to initiate the sequence. After a variable time (intertrial interval: 2000–3000 ms), a red laser projected a fixation spot under the object. If the animal maintained its gaze inside the electronically-defined fixation window (+/−2.5 deg) for 500 ms, the object was illuminated. Following a variable delay (900–1100 ms), a visual GO cue (dimming of the laser) instructed the monkey to lift the hand from the resting position, and reach, grasp, lift and hold the sphere for a variable interval (holding time: 500–900 ms). Whenever the monkey performed the whole sequence correctly, it received a drop of juice as reward. During the task, we measured both the time between the go-signal and the onset of the hand movement (reaction time), and the time between the start of the movement and the lift of the object (grasping time).

**Experimental protocol.** To address the influence of single-pulse TMS on neuronal activity, we interleaved trials with (50% of the trials) and without (50% of the trials) stimulation, and applied single magnetic pulses in different task epochs. These TMS pulses were applied in blocks of either low (60% rMT) or high (120% rMT) intensity TMS, randomly interleaved with no-stimulation trials. In both monkeys, we applied single-pulse TMS at two different task epochs: at light onset above the object, and immediately (within 2 ms) after the onset of the hand movement. In monkey P, we also run an experimental control to investigate the effect of TMS on task-related activity. For this, we applied high and low-intensity TMS at both hand lift and at hand lift + 400 ms. Across multiple experimental sessions, we measured the TMS-induced changes in the spiking activity of individual neurons located in a region under the center of the coil extending 7 mm in the antero-posterior direction and 3 mm in the medio-lateral direction (Supplementary Figs. 3 and 4).

**Data analysis.** All data analyses were performed in MATLAB (MathWorks, Massachusetts, USA). For the high- and low stimulation trials, the neural activity was aligned on the TMS pulse. Also, for comparison, the no-stimulation trials were aligned on the same time bin (either light onset or hand lift). Net neural responses were then calculated as the average firing rate recorded after TMS minus the baseline (spike rate calculated from the mean activity of the cell in the 800 ms interval preceding TMS).

We created line and raster plots comparing the average response (spikes/sec) of every cell during no stimulation, low stimulation and high stimulation trials at different task epochs. A bin-by-bin analysis (two-sided Wilxocoxon ranksum test; bin size: 20 ms) comparing the spike rate in high stimulation bins with that in the corresponding no-stimulation bins was performed to classify the neurons according to their functional properties (excitatory TMS effect, inhibitory TMS effect, or combination of excitatory and inhibitory TMS effects). Similarly, we also ran a two-sided Wilxocoxon ranksum test to identify those neurons with significant task-related activity. For this, we first aligned the neural activity on the lift of the hand and then compared the cell's responses obtained during the delay (−200–0 ms before the lifting of the hand) and the reaching-to-grasp phases of the task (10–410 ms after lifting the hand/TMS event).

To determine the significance of the TMS-evoked effect in individual cells, we compared the cell responses observed between 10 and 50 ms post-TMS in the high stimulation condition to those in the no-stimulation condition (two-sided Wilxocoxon ranksum test). A two-way ANOVA was performed to quantify the interaction between the factors *stimulation* and *task epoch*. To estimate the spatial spread of the observed TMS effect, we calculated the average response of all neurons recorded at each recording position during high-, low- and no-stimulation trials. For each recording grid position, we determined the significance of the TMS-evoked effect by performing a Wilcoxon ranksum test comparing the average spike rate across all neurons recorded at that position. In this analysis, we compared the bin showing the maximum response to stimulation (high stimulation trials) to the corresponding bin in the no stimulation trials (bin size: 20 ms). Next, we tested whether the TMS-evoked effect differed significantly between the center of stimulation (i.e., the recording position with the highest TMS effect) and four neighboring positions located 1 mm away (+1 mm anterior, +1 mm posterior, +1 mm medial and +2 mm medial to the center) using a three-way nested ANOVA with position and TMS condition (high versus no stimulation) as factors, and cells as nested factors. To normalize the activity across all neurons showing task-related activity, we used the *normr* function in Matlab.

Finally, we measured the behavioral effect of TMS by directly comparing the reaction times and grasping times obtained for the high stimulation and no stimulation conditions with single-pulse TMS applied at light onset and at the onset of the hand movement. To adjust for multiple testing, we applied the Bonferroni correction.

**Reporting summary.** Further information on research design is available in the Nature Research Reporting Summary linked to this article.

## Data availability
All datasets are available at Dryad (https://doi.org/10.5061/dryad.g54381n). Additional modified scripts can be accessed upon request. Similarly, all simNIBS parameters and brain volumes are available from the corresponding authors on request.

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

## Acknowledgements

This work was supported by Fonds voor Wetenschappelijk Onderzoek Vlaanderen, Odysseus (G.0007.12, G.0C51.13N), and Program Financing (PFV10/008). We would like to thank Axel Thielscher for providing the coil files and Wouter Depuydt for his assistance with the use of simNIBS. We also thank Stijn Verstraeten, Christophe Ulens, Piet Kayenbergh, Gerrit Meulemans, Marc De Paep, Astrid Hermans and Inez Puttemans for their technical contributions, and Steve Raiguel for comments on a previous version of this manuscript.

## Author contributions

R.M.C. and D.M. designed and performed the experiments and analyzed the data. A.M. performed the simNIBS simulations. R.M.C., D.M. and J.P. wrote the manuscript in consultation with A.M.

## Additional information

**Competing interests:** The authors declare no competing interests.

