## [Transparent Peer Review File · Nature Communications]

Reviewers' comments:

Reviewer #1 (Remarks to the Author):

This paper examines single-unit activity in two awake rhesus monkeys while receiving single-pulse transcranial magnetic stimulation (TMS). TMS pulses are applied at two phases of a reach and grasp task (at onset of visual go-signal or at onset of hand movement) at two TMS intensities (60% and 120% of motor threshold) over parietal cortex, of which a good fraction of neurons show movement-related activity in the movement phase. Neuronal activity following TMS is compared to a no TMS baseline. The results show enhanced spikes in the first 50ms after TMS for high but not low intensity TMS (sometimes followed by an inhibition of activity and then a recurrent spike burst). Intriguingly, even in high intensity TMS trials, the local spread of activity is very small (often covering only a few contacts), and not in alignment with modelling of the current distribution. Finally, in contacts where both TMS induced spikes and movement-related activity is observed (high intensity TMS), TMS suppresses movement-related activity, which goes along with a prolongation in movement time.

This is an intriguing report on TMS effects in a unique model (very few animal data available). I am not an animal researcher but have experience with TMS in healthy human participants in whom neural responses to TMS at the chosen high intensity level are usually substantial (both in terms of EEG and BOLD signal), in stark contrast with the reported weak effects. Hence, I have concerns that something in the experimental setting has gone wrong. The reasons for the discrepancy between the present and past TMS-EEG or TMS-fMRI findings would need to be better explained and accounts based on methodological issues ruled out.

Specific comments

1) There are numerous papers today on recording of electrophysiological signals (evoked potentials and brain oscillations) and BOLD signals in the human brain, all showing substantial neural responses. The introduction refers to some of these studies but many citations are only covering the seminal method papers, not the many ensuing original papers. The text is set up to contrast the model of current distribution (massive spread) against the present data (weak spread). However, what about the many TMS-EEG and TMS-fMRI findings? One wonders how to explain the discrepancy between the present and previous data, in particular because the present result seem the odd one out. Also, only covering the seminal methods papers does miss some of the current open questions, to which the present data set could have contributed. For instance, there is TMS-EEG work examining the effects of single-pulse TMS on brain oscillations (not cited). The presented data set hints at some recurrent patterns of activity (Fig 2B, with an interval of 100ms between spike trains, i.e. at an alpha frequency), but the analysis window is too short to examine its oscillatory nature (cut at 150ms, should ideally go to 1000ms). In sum, I am concerned about the quality of the main result on the one hand (not convinced there is not a problem, see point 2 below), while on the other hand I feel the data are underexploited in other dimensions (e.g. not covering open questions). This also leads to a rather descriptive paper, without much embedding in current state-of-the-art and theory (e.g. on mechanism of action of TMS).

2) I am puzzled by the weak effects. Could this be explained by methodological issues? I am inferring from the text that coil placement was on the scalp, i.e. not hindered by the recording chamber? If this is the case, could the electrode position be too remote from the stimulation site to pick up the bulk of TMS-induced activity? What was the distance of the recording site to the centre of the coil? Also, the method section describes an artefact-elimination procedure. Could this have eliminated neural signal as well (together with the artifacts)? Moreover, was TMS intensity be determined with or without the implanted electrodes? If without, actual TMS intensity may be much smaller due to enhanced distance of the coil from the scalp by the interposed recording chamber. If methodological issues can be ruled out, what is then the reason for the discrepancy? Could parts of this be explained by stimulation of inhibitory interneurons versus excitatory neurons

that lead to differential effects on spikes (e.g. reduced) versus evoked potentials in EEG?

Minor

3) Page 3, line 66: "PFG" seems the first use of the acronym, without introduction?

4) It would be nice if text would refer in ascending order to Figures (now referring to Figure 1b before Figure 1a).

Reviewer #2 (Remarks to the Author):

This is an important manuscript that will be of broad interest to many cognitive neuroscientists. It tells us something about the parietal cortex. However, most crucially it reveals very important aspects of the effect of a widely used minimally invasive brain stimulation tool, transcranial magnetic stimulation (TMS), on neural activity. As anyone who has ever held a TMS coil and searched for a subject's primary motor cortex will attest, TMS appears to exert an effect that is, in many ways, small and relatively focal. Its effects however, are often assumed to be quite the opposite because of various TMS field modelling tools suggest that this is the case. Romero and colleagues here report real neurophysiological data from non-anaesthetized animals that demonstrate the spatial and temporal nature of TMS effects and their interaction with endogenous activity in the same regions that is related to task performance. Surprisingly nothing quite like this information is available in the field and so this is an exceptionally useful resource that really ought to be referenced by any researcher using TMS. I have only some minor comments.

1 P6 When the relatively limited spatial spread of the effect is discussed it might be useful to explain two facts to the reader. First, the size of the coil is, at 25mm, much smaller than any that is used in human experiments; in human research probably the smallest commonly used coils are 50mm in size. The coil size is not, however, mentioned until the Methods on page 14. Some discussion on page 6 and / or in the Discussion about how the authors' coil and human TMS coils differ from and resemble one another might be useful. It might be helpful to explain this in the context of the differences in brain size and skull thickness between the human and macaque species. Some more discussion of these issues somewhere in the manuscript might increase the use that the widest group of readers can make of the authors' findings.

2 p6 Second, it might be useful to explain what exactly it is that the authors treat as the centre of the coil. I think that it is the centre of the join between the two wings of the figure-of-8. This is usually the point at which the magnetic field intensity is highest. However, a position just anterior to this is the place where the spatial differential of the magnetic field is highest and this was often assumed to be important in the past; for example this argument was made even by the inventor of the modern TMS coil (Barker, 1999; *The history and basic principles of magnetic nerve stimulation. Electroencephalography and Clinical Neurophysiology*, 3-21.) in his figure 16. The spatial differential of the field, however, is probably most important when there is a possibility of setting up a potential difference along the length of an axon as is the case in the peripheral nervous system. However, it is of some consequence in some brain areas because we know that coil orientation determines the likelihood of eliciting a motor evoked potential (MEP) when the primary motor cortex is stimulated. This is presumably because some orientations of the coil are more likely to affect the descending fibres as they leave the motor cortex. Romero's and colleagues' findings suggest that the field intensity is probably most critical in driving cortical neural activity in other situations.

3 The effects that TMS of parietal area PFG has on reaching are interesting but perhaps it should be noted that while they may be the first examinations of reversible effects restricted to PFG there are other demonstrations that permanent lesions of PFG and the rest of the inferior parietal lobule (Faugier-Grimond et al., 1985, *Experimental Brain Research*) or PFG (7ab) and adjacent posterior

inferior parietal lobule (7a/PG) (Rushworth et al., 1997; Experimental Brain Research) disrupt reaching.

4 p11, paragraph 2 The effects of parietal cortical TMS on movements in the context of visual perturbation was also reported by Glover et al., (Journal of Cognitive Neuroscience, 2005).

Reviewer #3 (Remarks to the Author):

In principle I commend the authors for the intention of this work. There are truly remarkable holes in our understanding of the MOA of TMS. I'm not sure sure this enhances that understanding in an unambiguous way.

I've work on modeling for a decade so when I'm rather confused by the entire TMS results section maybe it's not just me. I don't follow how the authors got to threshold, in absolute terms what was the e field and what Current was applied to the coil model (compared to that applied in each animal) and what 83% means in their thought process. The image shows diffuse current flow (cm). So have every other model and what's with best case assumptions (no coil movement. No consideration of inter animal anatomy). To their credit the authors run a model (though using a reduced open source software) but they can't square the model with what they claim experimentally. For an empirical effort maybe we can leave it at that but for a paper claiming to address MOA this seems not enough. Nor am I the least convinced the authors claim that "extreme" focality is supported by prior efforts. First I think that cherry picking the literature and if it was, then this paper is not novel.

I was a little uneasy with the broad use of colloquial and descriptive language "actual spread?". "Direct consequence of Neuromodulation" as opposed to what (what is controlled by intensity reduction)... "clear"... "barely"... "marginally"... "noticed"... "seemingly extraordinary"... I know we all fall back on such phrases but they appear often in this paper, making it seem like the story has an agenda. and the abstract has just one number the actual meaning of which is not clear.

Studies in man run many additional informative experiences from varied pulse intervals to coil rotations. A lot could have been learned using these variations, knows to profoundly impact outcome in man, when doing animal studies

The title (which includes a spelling mistake) talks about neuronal basis. I don't see hard insight on neuronal basis in the abstract and conclusion. Which neurons are activated? Which synapses or pathways? How can this be related to rTMS if at all? What did this prove about the neural basis (outside of spatial targeting for which I note my concerns) which was unexpected, important, and quantitative compared to the literature?

I appreciate this criticism is direct and harsh, but I hope found by the authors to be constructive as it is intended.

Tms would induce current in wires leading to direct electrical stimulation at the electrodes?

Reviewer #1 (Remarks to the Author):

This paper examines single-unit activity in two awake rhesus monkeys while receiving single-pulse transcranial magnetic stimulation (TMS). TMS pulses are applied at two phases of a reach and grasp task (at onset of visual go-signal or at onset of hand movement) at two TMS intensities (60% and 120% of motor threshold) over parietal cortex, of which a good fraction of neurons show movement-related activity in the movement phase. Neuronal activity following TMS is compared to a no TMS baseline. The results show enhanced spikes in the first 50ms after TMS for high but not low intensity TMS (sometimes followed by an inhibition of activity and then a recurrent spike burst). Intriguingly, even in high intensity TMS trials, the local spread of activity is very small (often covering only a few contacts), and not in alignment with modelling of the current distribution. Finally, in contacts where both TMS induced spikes and movement-related activity is observed (high intensity TMS), TMS suppresses movement-related activity, which goes along with a prolongation in movement time.

This is an intriguing report on TMS effects in a unique model (very few animal data available). I am not an animal researcher but have experience with TMS in healthy human participants in whom neural responses to TMS at the chosen high intensity level are usually substantial (both in terms of EEG and BOLD signal), in stark contrast with the reported weak effects. Hence, I have concerns that something in the experimental setting has gone wrong. The reasons for the discrepancy between the

present and past TMS-EEG of TMS-fMRI findings would need to be better explained and accounts based on methodological issues ruled out.

Specific comments

1) There are numerous papers today on recording of electrophysiological signals (evoked potentials and brain oscillations) and BOLD signals in the human brain, all showing substantial neural responses. The introduction refers to some of these studies but many citations are only covering the seminal method papers, not the many ensuing original papers. The text is set up to contrast the model of current distribution (massive spread) against the present data (weak spread). However, what about the many TMS-EEG and TMS-fMRI findings? One wonders how to explain the discrepancy between the present and previous data, in particular because the present result seem the odd one out. Also, only covering the seminal methods papers does miss some of the current open questions, to which the present data set could have contributed. For instance, there is TMS-EEG work examining the effects of single-pulse TMS on brain oscillations (not cited). The presented data set hints at some recurrent patterns of activity (Fig 2B, with an interval of 100ms between spike trains, i.e. at an alpha frequency), but the analysis window is too short to examine its oscillatory nature (cut at 150ms, should ideally go to 1000ms). In sum, I am concerned about the quality of the main result on the one hand (not convinced there is not a problem, see point 2 below), while on the other hand I feel the data are underexploited in other dimensions (e.g. not covering open questions). This also leads to a rather descriptive paper, without much embedding in current state-of-the-art and theory (e.g. on mechanism of action of TMS).

Reply:

The reviewer is puzzled by the discrepancy between our single-cell data and previous studies using TMS-EEG and concurrent TMS-fMRI. We have several answers to this comment:

- First, we would like to point out that the neural effects we measured were not weak at all, they were actually very strong, reaching more than 60 spikes/sec in the center (also see response to comment 2). In other words, we did not observe a 'weak spread' but rather a highly focal area of strong neuronal activity induced by TMS.
- We have now included several references to previous work on TMS-EEG in the revision. Moreover, we also analyzed the spike oscillations induced by TMS (new Supplementary Figure 7, described on p. 7 and discussed on p.11). Interestingly, TMS induced an increase in power in the low frequencies (below 5 Hz), both in the center and at a distance from the center of stimulation. Therefore, TMS induced oscillatory activity at a distance (even when the neurons did not increase their average firing rate), and this could be detected by EEG. We believe these additional data represent important information to reconcile our single-cell results to TMS-EEG data. That is, the highly focal TMS-induced spiking activity is dissociable from oscillatory activity, detectable by EEG and which spreads remotely to interconnected areas.
- We know from previous fMRI-single-cell experiments in monkeys (in which we recorded neural activity in an extensive fMRI activation in parietal cortex, Van Dromme et al., 2015, Neuroimage) that large fMRI activations can be due to very small clusters of neurons being

actually activated. Therefore, it is possible that the very focal neural activation we measured would appear as a relatively extended effect in fMRI.

Text:

p. 7 *'The TMS pulse not only influenced the average firing rate of the neurons under the coil, but also induced oscillatory single-unit activity. We performed a spectral analysis on the single-unit activity using a Hanning-tapered Fourier transformation, and observed significantly higher low-frequency oscillatory activity after high-intensity TMS (compared to no TMS, Supplementary Figure 7). However, this increase in power in the lower frequencies (below 10Hz) was also present outside the center of stimulation. Hence, single-pulse TMS induced low-frequency oscillatory activity across a region of cortex that was much wider than the area where we measured significant increases in single-unit firing rate.'*

p.11 *'Our results may also be surprising in the light of numerous TMS-EEG studies, which showed oscillatory activity in the EEG induced by single-pulse TMS (refs, e.g. Rosanova et al., 2009; Rogasch et al., 2015; Fecchio et al., 2017). However, at least three possible factors could explain this apparent discrepancy. Firstly, the EEG signal reflects mainly synaptic activations (Ilmoniemi and Kicic, 2010), whereas we performed extracellular recordings of action potentials in individual neurons. Secondly, we recorded the effect of single-pulse TMS in parietal cortex, and these parietal activations can certainly activate remote areas, e.g. in frontal cortex (as shown with electrical microstimulation during fMRI by Premereur et al., 2015), which could be detected by EEG. Finally, despite the absence of significant TMS-induced spiking responses at recording positions located 2 mm or more from the center of stimulation, TMS did induce strong (mainly low-frequency) oscillations in the firing rate of these neurons, which may also contribute to the EEG signal. It is therefore worth noting that the spatial spread of TMS-induced spiking activity is clearly dissociable from TMS-induced oscillatory activity, which spreads more remotely. Overall, it is most likely impossible to identify the exact location of a cortical activation on a millimeter scale with EEG, as we did in our single-cell recording experiments. Thus, concurrent EEG-TMS recordings would have probably shown a TMS-evoked potential across a large part of parietal cortex. A similar reasoning may apply to the apparent discrepancy between our results and TMS-fMRI studies (Bestmann et al., 2008; Blankenburg et al., 2010; Leitão et al., 2015). A previous study combining fMRI and single-cell recordings in parietal cortex of monkeys (Van Dromme et al., 2015) has demonstrated that very restricted clusters of active neurons can drive an extensive fMRI activation.'*

2) I am puzzled by the weak effects. Could this be explained by methodological issues? I am inferring from the text that coil placement was on the scalp, i.e. not hindered by the recording chamber? If this is the case, could the electrode position be too remote from the stimulation site to pick up the bulk of TMS-induced activity? What was the distance of the recording site to the centre of the coil? Also, the method section describes an artefact-elimination procedure. Could this have eliminated neural signal as well (together with the artifacts)? Moreover, was TMS intensity be determined with or without the implanted electrodes? If without, actual TMS intensity may be much smaller due to enhanced distance of the coil from the scalp by the interposed recording chamber. If methodological issues can be ruled out, what is then the reason for the discrepancy? Could parts of this be explained by stimulation of inhibitory interneurons versus excitatory neurons that lead to differential effects on spikes (e.g. reduced) versus evoked potentials in EEG?

Reply:

Again, we would first want to clarify that the neural effects we measured were not weak at all. For example, TMS induced a response which peaked at 64 spikes/sec at the center of stimulation (Fig 3D). Our main result is that TMS evokes a very focal activation (i.e. narrow spatial spread). However, to exclude the possibility that we stimulated near the threshold to activate the neurons, we ran a control experiment in which we stimulated at 100% of the resting motor threshold (i.e. 20% less than in our experiments, Supplementary Figure 3). The neural activation elicited by stimulation at 100% rMT was virtually identical to the one measured with the standard intensity (120% rMT), indicating that our standard intensity was clearly sufficient and at ceiling level for neuronal activation.

Secondly, we already mentioned in the text that our TMS coil was located 15 mm from the surface of the brain, which is comparable to experiments in humans (p.15). We acknowledge that the TMS-induced artefact lasting 8 ms may have obscured a small number of spikes. However, the large majority of the TMS effects lasted 40 ms or more, which makes it extremely unlikely that we missed a large part of the TMS-evoked response. We also clarified that the rMT was determined over M1 of the contralateral hemisphere with the recording chamber in place (but without inserted electrode), at a distance similar to the recording experiments. Please note we have not used an implanted electrode system. To clarify the apparent mismatch between EEG and our single-cell measurements, we now discuss several factors that could explain this:

- EEG detects mainly synaptic activity whereas we recorded action potentials.
- EEG can also detect remote activations (for example in frontal cortex) induced by parietal activation. We did not record in distant areas connected to parietal cortex.
- Importantly, TMS did induce low-frequency oscillations, even at recording sites far (more than 2 mm) away from the center of stimulation, which could be detected by EEG and reconciles our findings with TMS-EEG literature.

Text:

p. 11 *'Our results may also be surprising in the light of numerous TMS-EEG studies, which showed oscillatory activity in the EEG induced by single-pulse TMS (refs, e.g. Rosanova et al., 2009; Rogasch et al., 2015; Fecchio et al., 2017). However, at least three possible factors could explain this apparent discrepancy. Firstly, the EEG signal reflects mainly synaptic activations (Ilmoniemi and Kicic, 2010), whereas we performed extracellular recordings of action potentials in individual neurons. Secondly, we recorded the effect of single-pulse TMS in parietal cortex, and these parietal activations can certainly activate remote areas, e.g. in frontal cortex (as shown with electrical microstimulation during fMRI by Premereur et al., 2015), which could be detected by EEG. Finally, despite the absence of significant TMS-induced spiking responses at recording positions located 2 mm or more from the center of stimulation, TMS did induce strong (mainly low-frequency) oscillations in the firing rate of these neurons, which may also contribute to the EEG signal. It is therefore worth noting that the spatial spread of TMS-induced spiking activity is clearly dissociable from TMS-induced oscillatory activity, which spreads more remotely. Overall, it is most likely impossible to identify the exact location of a cortical activation on a millimeter scale with EEG, as we did in our single-cell recording experiments. Thus, concurrent EEG-TMS recordings would have probably shown a TMS-evoked potential across a large part of parietal cortex. A similar reasoning may apply to the apparent discrepancy between our results and TMS-fMRI studies (Bestmann et al., 2008; Blankenburg et al.,*

2010; Leitão et al., 2015). A previous study combining fMRI and single-cell recordings in parietal cortex of monkeys (Van Dromme et al., 2015) has demonstrated that very restricted clusters of active neurons can drive an extensive fMRI activation.'

p. 17 'In these experiments, the TMS coil was handheld over the primary motor cortex of the hemisphere contralateral to the hemisphere in which we recorded, at a distance of approximately 15 mm from the surface of the brain, similar to the distance in the recording experiments. The parietal recording chamber was implanted but no electrode was inserted in the brain.'

Minor

3) Page 3, line 66: "PFG" seems the first use of the acronym, without introduction?

Reply: PFG is not an acronym, it is the name for an area in the parietal convexity. Therefore, we cannot explain it further in the text.

4) It would be nice if text would refer in ascending order to Figures (now referring to Figure 1B before Figure 1A).

Reply: Thank you for spotting this. We have now corrected this.

Reviewer #2 (Remarks to the Author):

This is an important manuscript that will be of broad interest to many cognitive neuroscientists. It tells us something about the parietal cortex. However, most crucially it reveals very important aspects of the effect of a widely used minimally invasive brain stimulation tool, transcranial magnetic stimulation (TMS), on neural activity. As anyone who has ever held a TMS coil and searched for a subject's primary motor cortex will attest, TMS appears to exert an effect that is, in many ways, small and relatively focal. Its effects however, are often assumed to be quite the opposite because of various TMS field modelling tools suggest that this is the case. Romero and colleagues here report real neurophysiological data from non-anaesthetized animals that demonstrate the spatial and temporal nature of TMS effects and their interaction with endogenous activity in the same regions that is related to task performance. Surprisingly nothing quite like this information is available in the field and so this is an exceptionally useful resource that really ought to be referenced by any researcher using TMS. I have only some minor comments.

1) P6 When the relatively limited spatial spread of the effect is discussed it might be useful to explain two facts to the reader. First, the size of the coil is, at 25mm, much smaller than any that is used in human experiments; in human research probably the smallest commonly used coils are 50mm in size. The coil size is not, however, mentioned until the Methods on page 14. Some discussion on page 6 and / or in the Discussion about how the authors' coil and human TMS coils differ from and resemble one another might be useful. It might be helpful to explain this in the context of the

differences in brain size and skull thickness between the human and macaque species. Some more discussion of these issues somewhere in the manuscript might increase the use that the widest group of readers can make of the authors' findings.

Reply:

We have clarified that we used a smaller coil (55mm of external diameter), proportional to the smaller size of the monkey's head (on p. 4), and we have discussed this more in detail on p. 12. However, these small coils are also used in humans in the context of twin-coil paired-pulse studies where effective connectivity between premotor-primary motor areas are investigated (Davare et al., 2010, Cattaneo and Barchiesi, 2011; Johnen et al., 2015), making our results useful for human TMS studies.

Text:

p. 12 'Finally, we used a smaller coil (D25; 55 mm of external diameter) compared to the standard TMS coils used in human experiments (D70). However, this smaller coil was proportional to the smaller size of the monkey's skull. Moreover, we obtained very similar intensity thresholds with the D70 and the D25 coils when testing individual finger movements in our monkeys (see Methods).'

2) p6 Second, it might be useful to explain what exactly it is that the authors treat as the centre of the coil. I think that it is the centre of the join between the two wings of the figure-of-8. This is usually the point at which the magnetic field intensity is highest. However, a position just anterior to this is the place where the spatial differential of the magnetic field is highest and this was often assumed to be important in the past; for example this argument was made even by the inventor of the modern TMS coil (Barker, 1999; The history and basic principles of magnetic nerve stimulation. *Electroencephalography and Clinical Neurophysiology*, 3-21.) in his figure 16. The spatial differential of the field, however, is probably most important when there is a possibility of setting up a potential difference along the length of an axon as is the case in the peripheral nervous system. However, it is of some consequence in some brain areas because we know that coil orientation determines the likelihood of eliciting a motor evoked potential (MEP) when the primary motor cortex is stimulated. This is presumably because some orientations of the coil are more likely to affect the descending fibres as they leave the motor cortex. Romero's and colleagues' findings suggest that the field intensity is probably most critical in driving cortical neural activity in other situations.

Reply:

We estimated the center of stimulation based on the MRI with a dummy TMS coil over parietal cortex, and implanted the recording chamber based on this MR image. In the text, we use the term 'center of stimulation' to refer to the grid position with the strongest TMS-evoked response (p. 6). It is interesting to note that in both monkeys the recording sites with the largest TMS-evoked responses were located slightly more anteriorly (2 mm in monkey 2, and at 1 and 2 mm anterior in monkey 1) in the recording grid. However, we are not confident enough about the MRI localization to conclude that the physical center of stimulation (the join between the two wings) and the actual center of stimulation (based on the neural responses) were offset with respect to each other (which

would suggest that the location where the spatial differential of the magnetic field is highest determines the neural effects). Therefore, we prefer not to discuss this in the text.

3) The effects that TMS of parietal area PFG has on reaching are interesting but perhaps it should be noted that while they may be the first examinations of reversible effects restricted to PFG there are other demonstrations that permanent lesions of PFG and the rest of the inferior parietal lobule (Faugier-Grimond et al., 1985, Experimental Brain Research) or PFG (7ab) and adjacent posterior inferior parietal lobule (7a/PG) (Rushworth et al., 1997; Experimental Brain Research) disrupt reaching.

Reply:

These references are now included in the text on p. 14.

4 p11, paragraph 2 The effects of parietal cortical TMS on movements in the context of visual perturbation was also reported by Glover et al., (Journal of Cognitive Neuroscience, 2005).

Reply:

This is also now included in the text on p. 14.

Reviewer #3 (Remarks to the Author):

In principle I commend the authors for the intention of this work. There are truly remarkable holes in our understanding of the MOA of TMS. I'm not sure sure this enhances that understanding in an unambiguous way.

1) I've work on modeling for a decade so when I'm rather confused by the entire TMS results section maybe it's not just me. I don't follow how the authors got to threshold, in absolute terms what was the e field and what Current was applied to the coil model (compared to that applied in each animal) and what 83% means in their thought process. The image shows diffuse current flow (cm). So have every other model and what's with best case assumptions (no coil movement. No consideration of inter animal anatomy). To their credit the authors run a model (though using a reduced open source software) but they can't square the model with what they claim experimentally. For an empirical effort maybe we can leave it at that but for a paper claiming to address MOA this seems not enough. Nor am I the least convinced the authors claim that "extreme" focality is supported by prior efforts. First I think that cherry picking the literature and if it was, then this paper is not novel.

Reply:

We have now clarified the result section where the E field and the 83% threshold are presented. (p.3)

Overall, it is important to note that we minimized the TMS coil movements by locking it with 2 acrylic rods solidly implanted on the monkey skull. As far as inter-animal anatomy variability is concerned, we ran an MRI before our TMS sessions and located the recording chamber over PFG in a similar location in both monkeys.

2) I was a little uneasy with the broad use of colloquial and descriptive language “actual spread?”. “Direct consequence of Neuromodulation” as opposed to what (what is controlled by intensity reduction)... “clear”... “barely”... “marginally”... “noticed”... “seemingly extraordinary”... I know we all fall back on such phrases but they appear often in this paper, making it seem like the story has an agenda. and the abstract has just one number the actual meaning of which is not clear.

Reply:

We also changed the manuscript in many places to avoid the use of colloquial language, and we have added a clarification in the abstract (‘an area of cortex measuring less than 2 mm in diameter’).

e.g. on p. 4, p. 6 (‘significant’ instead of ‘clear’), p. 10 (‘the focality of the TMS effect’ instead of ‘the seemingly extraordinary focality ...’), etc.

3) Studies in man run many additional informative experiences from varied pulse intervals to coil rotations. A lot could have been learned using these variations, knows to profoundly impact outcome in man, when doing animal studies.

Reply:

We thank the reviewer for giving us the opportunity to present additional data. We have now added the results of another control experiment, in which we rotated the coil 180 degrees, so that the induced current ran anterior-to-posterior (AP instead of PA). Consistent with studies in humans, we observed that the neural activation was different, which is likely due to different neural populations being recruited by the TMS coil with each different orientation, and significantly smaller with the AP coil orientation compared to PA (Supplementary Figure 2).

Text:

p. 5 ‘In 15 neurons showing an effect of TMS, we inverted the coil so that the current flow was anterior-posterior (AP) instead of posterior-anterior (PA), and recorded the TMS-evoked activity under those conditions. Although we could elicit a TMS-evoked burst of activity with an inverted coil, the magnitude of this response was significantly smaller than that recorded with the standard orientation ($p = 0.04$, Wilcoxon, Supplementary Figure 2), consistent with observations in human volunteers that PA vs. AP coil orientations can recruit underlying neural populations differently (Di Lazzaro et al., 2001; Hamada et al., 2014).’

4) The title (which includes a spelling mistake) talks about neuronal basis. I don’t see hard insight on neuronal basis in the abstract and conclusion. Which neurons are activated? Which synapses or pathways? How can this be related to rTMS if at all? What did this prove about the neural basis

(outside of spatial targeting for which I note my concerns) which was unexpected, important, and quantitative compared to the literature?

Reply:

We agree with the referee and have changed the title to 'Neural effects of Transcranial Magnetic Stimulation at the single-cell level', which is more neutral.

5) I appreciate this criticism is direct and harsh, but I hope found by the authors to be constructive as it is intended. TMS would induce current in wires leading to direct electrical stimulation at the electrodes?

Reply:

We addressed this comment in the discussion.

Text:

p. 12 'Two technical factors may have influenced our findings. TMS may have induced a current in the microelectrode under the coil, causing direct electrical stimulation and consequently neuronal activation around the tip of the electrode. However, we did not observe such a neuronal activation at a distance of merely one millimeter from the center of stimulation (Figure 3), although the microelectrode was undoubtedly still in the magnetic field under the TMS coil. Therefore, it is extremely unlikely that the TMS pulse induced electrical stimulation at the tip of the electrode.'

References (articles not included in the manuscript):

Cattaneo L. & Barchiesi G. Transcranial magnetic mapping of the short-latency modulations of corticospinal activity from the ipsilateral hemisphere during rest. *Front. Neural Circuits*, **5**, 14 (2011).

Johnen V.M., Neubert F-X, Buch E.R., Verhagen L., O'Reilly J.X., Mars R.B. & Rushworth M.F.S. causal manipulation of functional connectivity in a specific neural pathway during behavior and at rest. *eLife*, **4**, e04585 (2015).

REVIEWERS' COMMENTS:

Reviewer #1 (Remarks to the Author):

This is an interesting study on the effects of TMS on neural activity in a rare model (non-human primates). All my concerns have been addressed. The results will be useful for the TMS community.

Reviewer #2 (Remarks to the Author):

I continue to think that this is an interesting manuscript. A key question ran through several of the reviews regarding the apparent discrepancy between the spatially restricted effects of TMS that the authors are reporting and previous demonstrations of more spatially widespread effects found using EEG and fMRI measurement. I think that the authors have done a good job of resolving the apparent conflict by explaining that while TMS-induced changes in spiking activity are spatially restricted, effects on oscillatory activity, which likely to reflect post-synaptic activity as opposed to just spiking activity, are more widespread. The authors have performed an important new analysis and reported the results. It might, however, be worth trying to re-read the manuscript as a naive reader would and emphasize in a few more places that the spatially selective effects are on spiking activity (for example, the abstract).

Reviewer #3

[Comments were contained in remarks to the editor]

Reviewer #2 (Remarks to the Author):

The authors have performed an important new analysis and reported the results. It might, however, be worth trying to re-read the manuscript as a naïve reader would and emphasize in a few more places that the spatially selective effects are on spiking activity (for example, the abstract).

Reply:

We have now clarified the effect on spiking activity both in the abstract and along the text.

Reviewer #3 (Remarks to the Author):

**In response to the explanation document that you kindly provided, Reviewer #3 indicates that he/she has no further objection to the publication of this manuscript although some of his/her concerns remain. We took the editorial decision to accept the manuscript in principle. However, we would like you to revise the manuscript in response to the following suggestion of Reviewer #3:*

"The authors propose the experimental results and modeling can be reconciled by "likely contained within E-field values between 93-100%." The current plots actually blurs 83-100 in a comparable red. Why not then replot the figure (or add additional panels) with a clear color cut off at 93%. My expectation is if the authors do this, the story remains a bit muddled in this regard."

We suggest that an additional panel or a new Supplementary Figure should be provided, along with some additional discussion as necessary.

Reply:

In response to the suggestion provided by the third reviewer, we have now included an extra Figure (Supplementary Figure 8), showing a 2 mm volume area, located immediately under the center of the coil and corresponding to that region where E-field values reached 95-100% of the maximum. According to our measurements, only the neurons located in this area showed a significant increase in their spiking activity.